# Social World Model-Augmented Mechanism Design Policy Learning

Xiaoyuan Zhang[1,2,3]∗   Yizhe Huang[1,2]∗   Chengdong Ma[1,3]   Zhixun Chen[4]   Long Ma[5]
Yali Du[6]   Song-Chun Zhu[2,1,3]†   Yaodong Yang[1,3]†   Xue Feng[2]†
[1]Institute for Artificial Intelligence, Peking University
[2]State Key Laboratory of General Artificial Intelligence, BIGAI, Beijing, China
[3] State Key Laboratory of General Artificial Intelligence, Peking University, Beijing, China
[4]The Hong Kong University of Science and Technology (Guangzhou)
[5]Center for Data Science, Academy for Advanced Interdisciplinary Studies, Peking University
[6]King's College London

## Abstract

Designing adaptive mechanisms to align individual and collective interests remains a central challenge in artificial social intelligence. Existing methods often struggle with modeling heterogeneous agents possessing persistent latent traits (e.g., skills, preferences) and dealing with complex multi-agent system dynamics. These challenges are compounded by the critical need for high sample efficiency due to costly real-world interactions. World Models, by learning to predict environmental dynamics, offer a promising pathway to enhance mechanism design in heterogeneous and complex systems. In this paper, we introduce a novel method named **SWM-AP** (**S**ocial **W**orld **M**odel-**A**ugmented Mechanism Design **P**olicy Learning), which learns a social world model hierarchically modeling agents' behavior to enhance mechanism design. Specifically, the social world model infers agents' traits from their interaction trajectories and learns a trait-based model to predict agents' responses to the deployed mechanisms. The mechanism design policy collects extensive training trajectories by interacting with the social world model, while concurrently inferring agents' traits online during real-world interactions to further boost policy learning efficiency. Experiments in diverse settings (tax policy design, team coordination, and facility location) demonstrate that SWM-AP outperforms established model-based and model-free RL baselines in cumulative rewards and sample efficiency.

## 1 Introduction

Mechanism design, the art of engineering incentive structures to guide self-interested agents towards desirable collective outcomes, underpins a vast array of societal functions, from resource allocation in digital economies and smart cities to the formulation of public policies [23]. Its profound significance lies in its potential to maximize social welfare, foster efficient cooperation, and resolve complex coordination problems in multi-agent systems [25]. However, traditional mechanism design paradigms often grapple with fundamental challenges inherent in real-world social systems. Chief among these is agent heterogeneity. Real-world populations consist of diverse individuals possessing persistent yet often unobservable latent traits (e.g., skills, preferences, risk attitudes), which critically influence their responses to incentives [21, 1]. Classical models frequently resort to simplifying assumptions of homogeneity or rely on unrealistic full information, leading to suboptimal or ineffective mechanisms.

---

∗Equal contribution
†Equal corresponding authors. Project website: https://sites.google.com/view/swm-ap/

39th Conference on Neural Information Processing Systems (NeurIPS 2025).

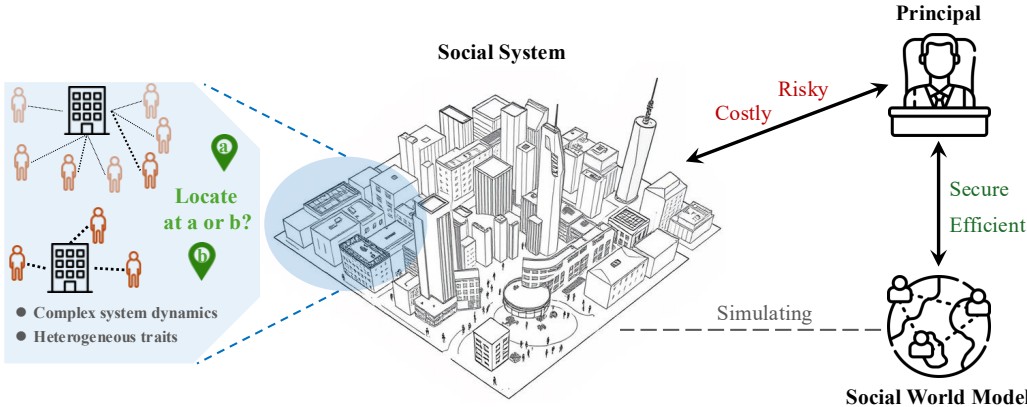

Figure 1: The AI mechanism designer (*principal*) makes decisions within a complex social system, exemplified by the facility location problem(see subsection 4.1 for details) shown on the left of the figure, which involves selecting the optimal site from potential locations (a or b). Directly interacting with the real social system is costly and risky. Such social systems typically present two major challenges. (1) Complex system dynamics, where individual interactions and environmental changes give rise to intricate and evolving system behaviors. (2) Heterogeneous agent traits, where individuals possess diverse latent preferences and needs (as illustrated in the magnified area on the left, where color intensity indicates preference for a facility, and the overall agent distribution also impacts the optimal location choice). Our proposed Social World Model aids the *principal* by simulating interaction in a secure and efficient manner.

Furthermore, these systems are characterized by complex system dynamics, which are difficult to capture using static, equilibrium-based analyses. Compounding these issues is the pervasive information asymmetry, where mechanism designers typically lack direct access to the crucial latent traits driving agent behavior. The advent of Artificial Intelligence, particularly Reinforcement Learning (RL), has ushered in a new era for mechanism design [39], offering unprecedented capabilities to develop adaptive and data-driven mechanisms capable of navigating increasingly complex and dynamic environments.

Model-Based Reinforcement Learning (MBRL) has emerged as a promising avenue for enhancing the efficacy and sample efficiency of mechanism design. By learning a model of the environment's dynamics, MBRL allows simulating trial-and-error exploration and counterfactual reasoning [22, 8], significantly reducing the reliance on costly real-world interactions, a critical advantage in high-stakes social systems. Despite this potential, the direct application of existing MBRL techniques to social mechanism design faces considerable hurdles. A primary limitation is the persistent neglect of agent heterogeneity. Many contemporary world models still treat agents as homogeneous entities or struggle to effectively represent and leverage their distinguishing latent traits. This oversight directly conflicts with the core need to design mechanisms tailored to the characteristics of the diverse agents [26, 12]. Moreover, modeling the intricate complexity of social interactions, encompassing cooperation, competition, and influence dynamics that can lead to highly non-linear and emergent system behaviors, poses a substantial challenge for standard world models [34], especially when individual agents are fundamentally driven by their underlying, unobservable traits.

To address these pressing challenges, we introduce a novel framework, named Social World Model-Augmented Mechanism Design Policy Learning (SWM-AP). Our approach, leveraging a model-based reinforcement learning paradigm, comprises two primary, interconnected components. The first is a sophisticated **Social World Model** (SWM), engineered to perform latent trait inference by unearthing agents' persistent hidden characteristics (e.g., skills, preferences) from their interaction trajectories in an unsupervised manner. It also learns trait-conditioned system dynamics, predicting how the social system evolves (i.e., state transitions and reward generation) as a function of these inferred traits and deployed mechanisms. The second core component is the **Mechanism Design Policy**. It is responsible for deploying optimal incentive mechanisms. This policy leverages the capabilities of SWM in two key ways: first, its prior mind tracker module conducts real-time inference of background agents' traits using the posterior mind tracker of SWM as the supervision signal;

second, it interacts with SWM's simulative environment to efficiently explore and refine its strategies. This synergy allows the Mechanism Design Policy to devise more adaptive, targeted, and ultimately effective incentive structures, aiming to maximize social welfare while minimizing the need for costly real-world samples, as shown in Figure 1.

The paper is structured as follows. We first review relevant literature on world models and mechanism design. Subsequently, we detail our SWM-AP framework with a theoretical analysis of algorithm feasibility. Through extensive numerical experiments across multiple scenarios, including facility location games, team optimization, and tax policy design, we validate our method's effectiveness. We conclude by summarizing contributions and proposing future directions for applying world models to real-world mechanism design research.

## 2 Related Works

### 2.1 Mechanism Design in Reinforcement Learning

The integration of Reinforcement Learning (RL) with mechanism design offers a powerful paradigm for dynamic systems, overcoming limitations of classical game-theoretic approaches tied to static equilibria and strict rationality. While foundational theories like Mirrlees' theory of optimal taxation [21] exist, they often struggle in dynamic, heterogeneous environments where agent preferences and capabilities evolve [4]. RL enables data-driven policy optimization via sequential interaction in complex settings [32, 18, 11, 2, 27]. However, contemporary RL-based mechanism design often oversimplifies agents as homogeneous, neglecting crucial cognitive traits (e.g., risk tolerance) that drive real-world decisions [30, 3]. While Inverse Reinforcement Learning (IRL) methods like 'Democratic AI' can infer latent preferences, they face scalability issues in co-training [16, 17]. Cognitive-aware RL advances this frontier: The $M^3RL$ framework [31] incorporates psychological states into policy adaptation but relies on explicit reward structures. Our work bridges these gaps by introducing a social world model that co-optimizes mechanism design. Many MARL approaches flat social structures for cooperative coordination [33, 24, 35, 15], while our hierarchical mechanism design guides self-interested agents.

### 2.2 World Model

Model-Based Reinforcement Learning methods learn dynamic models to guide policy optimization, reducing sample complexity while maintaining performance. The learned dynamic models fall into two categories. First is enhancing model-free methods with the learned model. Model-enhanced methods include MBPO [14], which trains the SAC or PPO algorithm using generated and real trajectories. Similar ideas have been extended to offline model-based RL settings [36]. Impressive advancements have also been made in learning dynamic changes in latent variable spaces [6, 5, 7, 8]. Furthermore, the application of transformers as world models [20] has demonstrated robust performance in humanoid robots [28, 38]. The second way is to use the model for planning. TD-MPC [10, 9] incorporates terminal value estimates for long-term reward estimates. Some existing works on multi-agent reinforcement learning employ world models to simulate the dynamics of multiple systems [37, 34, 19]. However, due to their lack of modeling complex social relationships among agents or explicit specification of agents' inherent attributes to simplify the problem, these approaches face challenges in deployment to real-world social environments. In contrast, our methodology simulates the dynamics of multi-agent systems characterized by individual heterogeneity and complex interaction relationships.

## 3 Method

In this paper, we propose a Social World Model-Augmented Mechanism Design Policy Learning **(SWM-AP)** approach, as illustrated in Figure 2 and Algorithm 1. In Subsection 3.1, we formally define the research problem through decision-making processes. Subsection 3.2 details the learning architecture of our SWM-AP approach, including the specifics of the Social World Model with its hidden trait inference tracker, and the training of the Mechanism Design Policy. Subsection 3.3 provides theoretical justification via ELBO derivation for our approach combining latent trait inference with world model learning, demonstrating the feasibility of jointly learning latent traits

and system dynamics, thereby supporting our approach of using trait inference to enhance the state prediction accuracy.

## 3.1 Problem Formulation

We formalize AI-driven mechanism design as an episodic Markov game between an institutional planner (*principal*) and a population of background agents. The *principal* operates as an algorithmic policy designer that dynamically deploys incentive mechanisms to optimize social welfare(the sum of all the background agents' returns). Each background agent maintains a fixed trait (such as skill or preference), which regulates the agent's response to the mechanism and shapes its behavior during interactions with other agents. These traits are private and unobservable to the *principal*, constituting a central challenge for adaptive mechanism design.

This problem can be succinctly summarized by the tuple $\langle N, \{\mathcal{M}_i\}, \mathcal{S}^{\text{obs}}, \mathcal{A}, P, R^{\text{soc}}, \gamma \rangle$. Here, $N$ is the number of background agents, each agent $i$ with a latent trait $m^i \in \mathcal{M}_i$. $\mathcal{S}^{\text{obs}}$ represents the *principal*'s comprehensive observation space, encompassing background agents' states $(s^1, ..s^N)$ that are visible to *principal* (like agents' locations) and global environment state $s^E$ (like the distribution of resources). The *principal* acts by selecting a mechanism policy $\pi \sim \Pi(\pi|s^{\text{obs}})$, while each background agent $i$ takes action following its policy $\phi_i(a^i|s^{\text{obs}}, \pi, m^i)$. The system transition function $P$ describes how observations evolve given current observations and the deployed mechanism policy. That is

$$s_{t+1}^{\text{obs}} \sim P(s_{t+1}^{\text{obs}}|s_t^{\text{obs}}, \pi_t). \tag{1}$$

The *principal* infers the system transition function $P$ and learns a mechanism policy $\pi$ to maximize the expected cumulative social welfare

$$\max_{\pi} \mathbb{E}\left[\sum_{t=0}^{\infty} \gamma^t r_t^{soc}\right], \tag{2}$$

where $r^{\text{soc}} = \sum_i r^i$ is *principal*'s reward, named the social welfare, $r^i = R^i(s^{\text{obs}}, a^1, ..., a^N)$ is the reward of background agent $i$, and $\gamma$ is the discount factor.

Thus, our formulation of AI-driven mechanism design presents two fundamental departures from classical mechanism design theory. 1) *Principal* has no prior knowledge of background agents' behavior patterns, which adaptively respond to the policies of *principal* and other background agents. *Principal* needs to conduct online inference of these patterns through interactions with them. 2) Agents exhibit heterogeneity, possessing diverse and persistent latent traits that individually shape their behaviors. Consequently, the *principal* must infer these distinct agent-specific traits and behavioral patterns from interaction trajectories, rather than relying on aggregated or homogeneous population models.

## 3.2 Social World Model-Augmented Mechanism Design Policy Learning

Our approach to AI-driven mechanism design problems within heterogeneous and dynamic multi-agent systems, termed Social World Model-Augmented Mechanism Design Policy Learning (**SWM-AP**), leverages a model-based reinforcement learning framework. The core of our method consists of two primary, interconnected components: a **Social World Model (SWM)** that learns the complex dynamics of state transition, and a **Mechanism Design Policy** that learns to deploy optimal incentive mechanisms. This overall architecture is designed to address the challenge of unobservable agent traits and to maximize social welfare with a minimized sample.

**Social World Model:** SWM is tasked with learning a comprehensive model of the state transition function. Specifically, given the current observation $s_t^{\text{obs}}$, the deployed mechanism $\pi_t$, and an estimate of the agents' latent traits $\hat{\mathbf{m}} = (\hat{m}^1, ..., \hat{m}^N)$, SWM learns to predict the next observation $s_{t+1}^{\text{obs}}$ and the immediate social welfare $r_t^{\text{soc}}$.

An important component of SWM is the Posterior Trait Tracker. Since the background agents' traits $\mathbf{m}$ are latent, accurate modeling of environment dynamics necessitates inferring these traits. The Posterior Trait Tracker is designed to infer these latent traits, $\hat{\mathbf{m}}_{\text{post}}$, by analyzing complete interaction trajectories $\tau = (s_0^{\text{obs}}, \pi_0, r_0^{\text{soc}}, \ldots, s_T^{\text{obs}}, \pi_T, r_T^{\text{soc}})$ collected during training. This module processes entire trajectory segments to capture long-term behavioral patterns indicative of the underlying traits. SWM, including its state prediction component, is then trained by minimizing the discrepancy

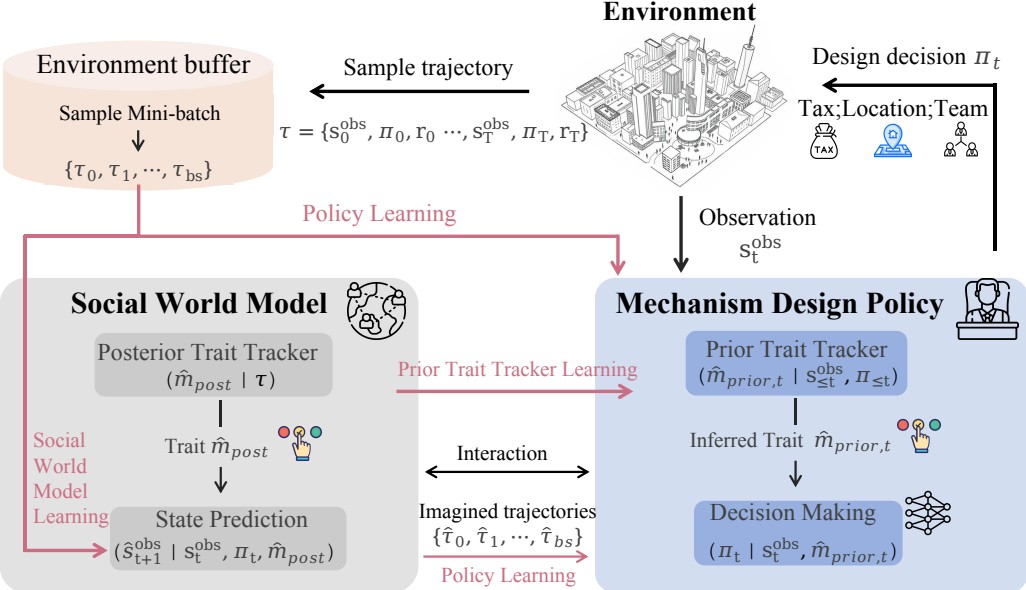

Figure 2: Algorithm diagram of SWM-AP. The **Social World Model (SWM)** utilizes a Posterior Trait Tracker to infer latent agent traits from full trajectories, which subsequently aid in predicting the future states of background agents. Concurrently, the **Mechanism Design Policy** employs a Prior Trait Tracker for real-time inference of agent traits based on partial history, informing its mechanism design. Interactions between the policy and the learned SWM, leveraging imagined trajectories, enhance sample efficiency for policy learning.

between its predicted future states and the actual observed states, utilizing these inferred traits $\hat{\mathbf{m}}_{\text{post}}$. The objective function of SWM can be expressed as:

$$J_{\text{SWM}}(\psi, \phi) = \mathbb{E}_{\tau \sim D} \left[ \sum_t \|\hat{s}_{t+1}^{\text{obs}}(s_t^{\text{obs}}, \pi_t, \hat{\mathbf{m}}_{\text{post}}) - s_{t+1}^{\text{obs}}\|_2^2 + c D_{KL} \left( p(\hat{\mathbf{m}}_{\text{post}}) || U(\mathbf{m}) \right) \right]. \quad (3)$$

Here, $J_{\text{SWM}}(\psi, \phi)$ represents the loss function for SWM with parameters $\psi$ and the Posterior Trait Tracker with parameters $\phi$. The expectation $\mathbb{E}_{\tau \sim D}$ is taken over trajectories $\tau$ sampled from a dataset $D$ of past experiences. $\hat{s}_{t+1}^{\text{obs}}$ is the next state observation predicted by SWM, conditioned on the current state $s_t^{\text{obs}}$, mechanism $\pi_t$, and the traits $\hat{\mathbf{m}}_{\text{post}}$ inferred by the Posterior Trait Tracker. $s_{t+1}^{\text{obs}}$ is the actual observed next state. The term $\| \cdot \|_2^2$ denotes the squared L2 norm (Euclidean distance), measuring the prediction error. $p(\hat{\mathbf{m}}_{\text{post}})$ is the probability output by the Posterior Trait Tracker, while $U(\mathbf{m})$ is the uniform distribution for every element in $\mathbf{m}$. $c$ is the regularization coefficient. This joint optimization allows SWM to learn environment dynamics that are conditioned on a rich understanding of agent traits.

**Mechanism Design Policy:** The Mechanism Design Policy, denoted as $\Pi(\pi_t|s_t^{\text{obs}}, \hat{\mathbf{m}}_{\text{prior}})$, is responsible for selecting and deploying incentive mechanisms $\pi_t$ to maximize the expected cumulative discounted social welfare, as defined in Equation 2. This policy is trained using PPO [29], interacting with the environment (either real or simulated by SWM) to gather experiences. The social welfare $r_t^{\text{soc}}$ serves directly as the reward signal for policy updates.

A critical challenge during policy deployment is that complete future trajectories are unavailable for the Posterior Trait Tracker. To address this, the policy component incorporates a **Prior Trait Tracker**. This module is trained to perform real-time inference of background agents' traits, $\hat{\mathbf{m}}_{\text{prior}}$, based on the historically observed partial trajectory up to timestep $t$, i.e., $(s_0^{\text{obs}}, \pi_0, \ldots, s_t^{\text{obs}})$. The Prior Trait Tracker is trained in a supervised fashion, typically by minimizing the discrepancy between its predictions $\hat{\mathbf{m}}_{\text{prior},t}$ and the "ground truth" traits $\hat{\mathbf{m}}_{\text{post}}$ inferred by the Posterior Trait Tracker from complete trajectories during offline training. For example, a common objective is to minimize a cross-entropy loss if traits are categorical or a mean squared error if traits are continuous, at each

step:

$$J_{\text{Prior}}(\xi) = \mathbb{E}_{\tau \sim D}\left[\sum_t L(\hat{\mathbf{m}}_{\text{prior},t}(s_{\leq t}^{\text{obs}}, \pi_{<t}, \mathbf{a}_{<t}), \hat{\mathbf{m}}_{\text{post}})\right]. \tag{4}$$

Here, $J_{\text{Prior}}(\xi)$ is the loss for the Prior Trait Tracker with parameters $\xi$. $L$ is loss function comparing the prior tracker's estimate at time $t$, $\hat{\mathbf{m}}_{\text{prior},t}$, which is based on observations up to $t$, with the more accurate posterior estimate $\hat{\mathbf{m}}_{\text{post}}$ derived from the full trajectory.

During online interaction (policy execution), the inferred trait $\hat{\mathbf{m}}_{\text{prior}}$ from the Prior Trait Tracker is fed as input to both the Mechanism Design Policy $\Pi$ (to inform its decision-making) and to SWM (when SWM is used for generating imagined trajectories). This allows the policy to adapt its mechanism design strategy dynamically based on its evolving understanding of the agents' latent traits. Furthermore, the policy interacts with the learned SWM to enhance sample efficiency. For instance, SWM can generate simulated rollouts under different candidate mechanisms, allowing the policy to be refined with significantly more data than direct environment interaction alone would permit. The Prior Trait Tracker's output can also be a probability distribution over possible traits, reflecting its prediction certainty, which the policy can leverage for more robust strategy learning [40]. This two-pronged approach, combining a world model that understands agent traits with a policy that leverages this understanding for real-time adaptation and efficient learning, forms the backbone of our method.

### 3.3 Theoretical Analysis

We derive the Evidence Lower Bound (ELBO) to provide a theoretical basis for unsupervised learning of latent agent traits from interaction data, within the episodic framework defined in subsection 3.1.

We denote the *principal*'s trajectory as $\tau = (s_0^{\text{obs}}, \pi_0, s_1^{\text{obs}}, \pi_1, \cdots, s_{T-1}^{\text{obs}}, \pi_{T-1}, s_T^{\text{obs}})$, the joint actions as $\mathbf{a}_t = (a_t^1, \cdots, a_t^N)$, and the joint traits as $\mathbf{m} = (m^1, \cdots, m^N)$. We assume the joint distribution of $\tau$ and $\mathbf{m}$ follows a generative process:

$$p(\tau, \mathbf{m}) = p(\mathbf{m})p(s_0^{\text{obs}})\prod_{t=0}^{T-1} p(\pi_t|s_{\leq t}^{\text{obs}})\int_{\mathbf{a}_t}\left(p(s_{t+1}^{\text{obs}}|s_t^{\text{obs}}, \pi_t, \mathbf{a}_t, \mathbf{m})\left(\prod_{i=1}^{N}\beta_i(a_t^i|s_t^i, m^i, \pi_t)\right)\right)d\mathbf{a}_t, \tag{5}$$

where $p(\mathbf{m})$ is the prior of the joint traits, and $\beta_i(\cdot|s_t^i, m^i, \pi_t)$ is the agent's fixed policy conditioned on its trait $m^i$ the deployed mechanism $\pi_t$.

The world model $p_\psi(s_{t+1}^{\text{obs}}|s_t, \pi_t, \mathbf{m})$ approximates the dynamics $\int_{\mathbf{a}_t}\left(p(s_{t+1}^{\text{obs}}|s_t^{\text{obs}}, \pi_t, \mathbf{a}_t, \mathbf{m})\left(\prod_{i=1}^{N}\beta_i(a_t^i|s_t^i, \mathbf{m}, \pi_t)\right)\right)d\mathbf{a}_t$. The policy of the *principal* $\Pi_\theta(\pi_t|s_{\leq t})$ provides $p(\pi_t|s_{\leq t})$. Thus, the likelihood is estimated as

$$p_{\psi,\theta}(\tau|\mathbf{m}) = p(s_0^{\text{obs}})\prod_{t=0}^{T-1}\Pi_\theta(\pi_t|s_{\leq t}^{\text{obs}})p_\psi(s_{t+1}^{\text{obs}}|s_t^{\text{obs}}, \pi_t, \mathbf{m}).$$

To generate new trajectories, we need to sample $\mathbf{m}$ from the posterior $p(\mathbf{m}|\tau)$ rather than the prior $p(\mathbf{m})$. However, $p(\mathbf{m}|\tau)$ is intractable, and we use the Posterior Trait Tracker $q_\phi(\mathbf{m}|\tau)$ to approximate it.

To maximize the log evidence $\log p(\tau)$, we need to maximize the ELBO:

$$\mathcal{L}_{\text{ELBO}}(\phi, \psi, \theta; \tau) = \underbrace{\sum_{t=0}^{T-1}\mathbb{E}_{q_\phi(\mathbf{m}|\tau)}[\log p_\psi(s_{t+1}^{\text{obs}}|s_t^{\text{obs}}, \pi_t, \mathbf{m})]}_{\text{state prediction likelihood}} \tag{6}$$

$$+ \underbrace{\sum_{t=0}^{T-1}\mathbb{E}_{q_\phi(\mathbf{m}|\tau)}[\log \Pi_\theta(\pi_t|s_{\leq t})]}_{\textit{principal}\text{ policy likelihood}} - \underbrace{D_{KL}(q_\phi(\mathbf{m}|\tau)||p(\mathbf{m}))}_{\text{regularization}}.$$

As we use the same *principal* policy $\Pi_\theta$ to collect real trajectories and generate simulated trajectories, the *principal* policy likelihood is always maximized for ELBO. We minimize Equation 3 to maximize

Equation 6 under assumptions that $p(s_{t+1}^{\text{obs}}|s_t^{\text{obs}}, \pi_t, \mathbf{m})$ is Gaussian and each element of $p(\mathbf{m})$ is uniform. Please check the derivation details in Appendix A.

---

**Algorithm 1** SWM-AP Learning framework

---

1: **Initialize:** Mechanism Design Policy $\Pi_\theta(\pi_t|s_t^{\text{obs}}, \hat{\mathbf{m}}_{\text{prior},t})$, Dynamic Model $M_\phi(\hat{s}_{t+1}^{\text{obs}}|s_t^{\text{obs}}, \pi_t, \hat{\mathbf{m}}_{\text{post}})$, Posterior Trait Tracker $q_\varphi(\hat{\mathbf{m}}_{\text{post}}|\tau)$, Prior Trait Tracker $p_\xi(\hat{\mathbf{m}}_{\text{prior},t}|H_t)$ where $H_t = (s_{\leq t}^{\text{obs}}, \pi_{<t}, \mathbf{a}_{<t})$, Environment, Model Datasets $D_{env}, D_{model}$
2: **for** $N Epochs$ **do**
3:     Collect real trajectories $\tau = (s_0^{\text{obs}}, \pi_0, r_0^{\text{soc}}, \dots, s_T^{\text{obs}}, \pi_T, r_T^{\text{soc}})$ in Environment using policy $\Pi_\theta$ and Prior Trait Tracker $p_\xi$. Store in $D_{env}$.
4:     Jointly train Posterior Trait Tracker $q_\varphi$ and Dynamic Model $M_\phi$ on dataset $D_{env}$, using objective based on Equation 3, implicitly training $q_\varphi$ to produce $\hat{\mathbf{m}}_{\text{post}}$.
5:     Train Prior Trait Tracker $p_\xi$ on dataset $D_{env}$, using objective based on Equation 4 to align $p_\xi(\cdot|H_t)$ with $q_\varphi(\cdot|\tau)$.
6:     Generate imagined trajectories $\hat{\tau} = (\hat{s}_0^{\text{obs}}, \pi_0, \hat{r}_0^{\text{soc}}, \dots)$ using Dynamic Model $M_\phi$, policy $\Pi_\theta$, and Posterior Trait Tracker $p_\xi$. Store in $D_{model}$.
7:     Optimize policy $\Pi_\theta$ using data from $D_{env}$ and $D_{model}$, maximizing objective Equation 2 using PPO on combined data.
8: **end for**
9: **Return:** Policy $\Pi_\theta$, SWM $p_\psi$

---

# 4 Experiments

## 4.1 Facility Location

We designed a facility location game to examine the effectiveness of the methodology, where we developed learning strategies to enhance the capability of higher-level agents in selecting optimal facility construction locations for the background populations of rule-based agents.

**Environment Setting:** In the facility location game, the mechanism designer is tasked with selecting appropriate facility locations for multiple agents distributed across a map. Different agents exhibit heterogeneous preferences regarding facility locations. Our approach achieves optimal mechanism design by learning dynamic mechanism design strategies to maximize the collective reward of multiple low-level agents. The environment is configured as a matrix where each agent maintains a fixed global position at the beginning of each round, representing their permanent residence in real-world scenarios. The mechanism designer determines facility locations each round with a fixed total number of facilities, where each location configuration influences the total visitation frequency of low-level agents to facilities. The reward is defined as the summation of visitation frequencies from low-level agents to facilities. This experimental setup corresponds to the classical facility location game in mechanism design theory. Specifically, we implement a configuration with five facilities and eight agents distributed across an $8 \times 8$ grid.

**Performance Analysis:** We evaluate our proposed **SWM-AP** method against several baselines: the model-based reinforcement learning algorithms Dreamer and MBPO, and the model-free RL algorithm PPO. The comparative results are presented in Figure 3. Figure 3a utilizes a dual y-axis plot to illustrate two key performance aspects: sample efficiency and final converged reward. On this plot, the circles, aligned with the right y-axis, represent the mean final reward ($\pm$ standard deviation) achieved by each algorithm upon convergence. The bars, corresponding to the left y-axis, indicate the number of training steps required for each method to reach a predefined performance target, specifically PPO's final converged reward. The results demonstrate that model-based methods generally exhibit superior sample efficiency compared to their model-free counterparts. Notably, our method not only achieves the highest sample efficiency, requiring the fewest training steps to meet the performance target, but also attains the highest final converged reward among all evaluated algorithms. Figure 3b and Figure 3c display the learning loss curves for system states and immediate rewards, respectively. We find that our SWM achieves more accurate predictions for both metrics when compared to other baselines.

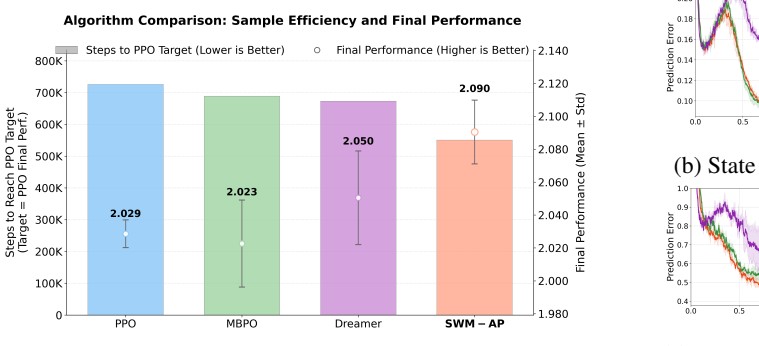

(a) Facility location: Sample efficiency and final performance

(b) State prediction loss

(c) Reward prediction loss

Figure 3: Facility location performance analysis. (a) Comparison of sample efficiency and final converged performance. (b) State prediction loss and (c) Reward prediction loss curves for our SWM compared to baselines.

## 4.2 Team Structure Optimization

Team structure optimization, where the team structure of background agents can be dynamically adjusted by *principal*, is another widely studied mechanism design problem. We conduct the experiments in AdaSociety [13], a highly customizable multi-agent environment supporting dynamic social relationships and heterogeneous agents with open-ended tasks. By controlling the relationships between background agents, *principal* aims to maximize the collective reward of background agents.

**Environment Setting:** The environment consists of an $8 \times 8$ grid with four types of basic resources (10 units each, positioned at the corners), where two basic resources can be converted into one advanced resource (valued at 5, compared to 1 for basic resources). Four agent types exist, each capable of producing a specific basic resource but unable to store it. Instead, they can store one other predefined resource type. Agents form teams of arbitrary size, with each agent restricted to one team, and incur a maintenance cost of $0.05(x - 1)$ per step, where $x$ is the team size. Agents produce resources matching their type and only store resources to earn rewards if a teammate can produce their storable types. In each episode, four background agents are initialized with randomly assigned types. *Principal* observes the current map without knowing agents' types, and then reassigns the team structure every 10 steps starting at step 5, aiming to maximize total group reward. Each episode lasts for 50 timesteps. The task challenges *principal* to optimize collective efficiency in a resource-constrained multi-agent system, where *principal* must balance production, storage dependencies, and team coordination costs.

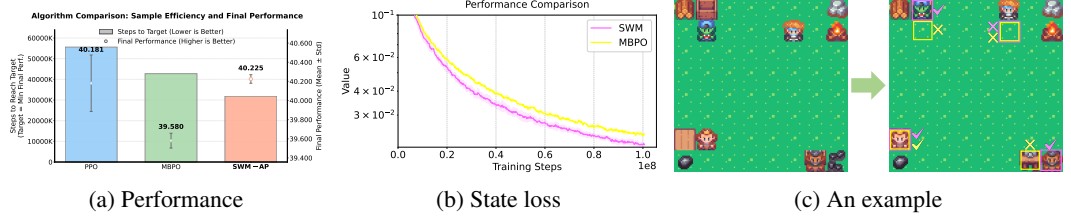

(a) Performance        (b) State loss        (c) An example

Figure 4: Performance comparison in team structure optimization

**Performance Analysis:** We evaluate our **SWM-AP** method against representative model-based and model-free RL baselines in the dynamic team structure optimization task within AdaSociety. The comparative results are presented in Figure 4. Figure 4a highlights key performance metrics: sample efficiency (bars, left y-axis), indicating training steps to reach a target performance, and final converged group reward (circles/points, right y-axis, with ± standard deviation). Consistent with findings in other domains, model-based approaches demonstrate superior sample efficiency. Notably, our method not only achieves the highest sample efficiency, requiring the fewest steps to reach the target, but also secures the highest final converged group reward, showcasing its effectiveness in optimizing team configurations under resource constraints and dynamic reassignments. Figure 4b

depicts the learning curves for the predictive components of the world model. These plots indicate that our SWM, which explicitly infers and uses agent traits to model team dynamics, achieves lower prediction errors compared to model-based baselines. Figure 4c illustrates an example of SWM and MBPO in this environment. Both algorithms take the current state (left sub-graph) as input and predict the location of each agent in the next timestep. These predictions are shown in the right sub-graph, with our SWM's prediction marked in pink and MBPO's in yellow. The right sub-graph also displays the actual location of the agent in the next timestep for comparison. Overall, SWM's predictions are more accurate. For instance, for the agent in the lower right corner, SWM correctly predicts that the agent will move to the grid on the right, where resources are located, while MBPO predicts that the agent will stay in its current position, where the manufacturing of advanced resources is taking place. This accuracy is likely because SWM has a better understanding of background agents' traits, enabling it to more precisely infer these agents' action plans.

### 4.3 Tax Adjustment

In this experimental setup, low-level agents are trained using reinforcement learning. As a result, the planner interacts with a continually adapting and improving population of agents, making the environment increasingly complex over time.

**Environment Setting:** In AI-Economist [39], background agents engage in activities such as collecting materials (specifically wood and stone) to construct houses in exchange for income. They can also trade resources on the market. Agents possess varying levels of skills in house construction, and their primary objective is to maximize individual utility. This utility is positively influenced by income but negatively affected by labor effort. Therefore, agents make decisions by considering several economic variables, including their construction skills, resource endowments, and applicable tax rates. These factors influence both their work and consumption choices. *Principal*, whose role is to design tax policies, seeks to enhance social welfare by balancing overall economic productivity with income equality. Notably, *principal* lacks visibility into the agents' specific construction skills. For this experiment, the environment consists of four low-level agents operating within a $25 \times 25$ map. Each episode spans 1000 time steps, while *principal* updates tax policies every 100 steps.

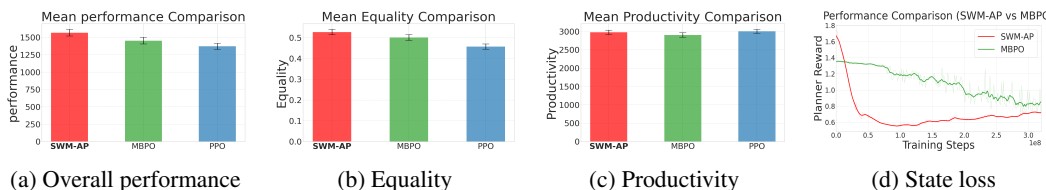

(a) Overall performance      (b) Equality      (c) Productivity      (d) State loss

Figure 5: Performance comparison in AI-Economist

**Performance Analysis:** We assess the effectiveness of our SWM-AP by comparing it with two baseline algorithms: the model-based MBPO and the model-free PPO. The evaluation is based on three social metrics recorded after 1,000 agent steps: *equality*, *productivity*, and their product, *equality × productivity*. The product metric serves as an indicator of overall social welfare within the simulated environment. The comparative results are presented in Figure 5. As shown, our method exhibits effective control over taxation, enabling *principal* to optimize social welfare. Notably, our method attains a level of productivity comparable to that of PPO while significantly enhancing equality. This suggests that our SWM is capable of promoting a more equitable society without compromising economic output. In contrast, MBPO yields suboptimal performance. Although it attempts to increase equality, this comes at the cost of a marked decline in productivity, ultimately resulting in lower overall social welfare relative to our method. We attribute the superior performance of our method to its ability to reduce state prediction error more efficiently during the early phases of training (as shown in Figure 5d). This improved predictive accuracy facilitates more effective *principal* optimization, thereby leading to better overall outcomes.

## 5 Conclusion

This paper proposes a social world model-augmented mechanism design policy learning method, named SWM-AP, which employs unsupervised learning to infer the hidden trait of background

agents, thereby enhancing the prediction of group dynamics and mechanism design policy learning. Experimental results in facility location, team structure optimization, and taxation setting tasks demonstrate that our method outperforms both model-based and model-free reinforcement learning approaches. Our approach inspires new directions for world model research towards more complex and realistic social world.

**Limitations and Future Work:** Current limitations include the scalability of SWM to extremely large-scale systems and the challenge of ensuring the direct interpretability of all inferred latent traits. Future work will focus on developing more scalable SWM architectures, potentially leveraging structured priors, and on enhancing trait interpretability through techniques like disentangled representation learning.

## Acknowledgments and Disclosure of Funding

This work is supported by the National Science and Technology Major Project (No. 2022ZD0114904).

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

# A ELBO Derivation

We denote the *principal*'s trajectory as $\tau = (s_0^{\text{obs}}, \pi_0, s_1^{\text{obs}}, \pi_1, \cdots, s_{T-1}^{\text{obs}}, \pi_{T-1}, s_T^{\text{obs}})$, the joint actions as $\mathbf{a}_t = (a_t^1, \cdots, a_t^N)$, and the joint traits as $\mathbf{m} = (m^1, \cdots, m^N)$. We assume the joint distribution of $\tau$ and $\mathbf{m}$ follows a generative process:

$$
p(\tau, \mathbf{m}) = p(\mathbf{m})p(s_0^{\text{obs}}) \prod_{t=0}^{T-1} p(\pi_t|s_{\leq t}^{\text{obs}}) \int_{\mathbf{a}_t} \left( p(s_{t+1}^{\text{obs}}|s_t^{\text{obs}}, \pi_t, \mathbf{a}_t, \mathbf{m}) \left( \prod_{i=1}^N \beta_i(a_t^i|s_t^i, m^i, \pi_t) \right) \right) d\mathbf{a}_t,
\tag{7}
$$

where $p(\mathbf{m})$ is the prior of the joint traits, and $\beta_i(\cdot|s_t^i, m^i, \pi_t)$ is the agent's fixed policy conditioned on its trait $m^i$ the deployed mechanism $\pi_t$.

The world model $p_\psi(s_{t+1}^{\text{obs}}|s_t, \pi_t, \mathbf{m})$ approximates the dynamics $\int_{\mathbf{a}_t} \left( p(s_{t+1}^{\text{obs}}|s_t^{\text{obs}}, \pi_t, \mathbf{a}_t, \mathbf{m}) \left( \prod_{i=1}^N \beta_i(a_t^i|s_t^i, \mathbf{m}, \pi_t) \right) \right) d\mathbf{a}_t$. The policy of the *principal* $\Pi_\theta(\pi_t|s_{\leq t})$ provides $p(\pi_t|s_{\leq t})$. Thus, the likelihood is estimated as

$$
p_{\psi,\theta}(\tau|\mathbf{m}) = p(s_0^{\text{obs}}) \prod_{t=0}^{T-1} \Pi_\theta(\pi_t|s_{\leq t}^{\text{obs}})p_\psi(s_{t+1}^{\text{obs}}|s_t^{\text{obs}}, \pi_t, \mathbf{m}).
$$

To generate new trajectories, we need to sample $\mathbf{m}$ from the posterior $p(\mathbf{m}|\tau)$ rather than the prior $p(\mathbf{m})$. However, $p(\mathbf{m}|\tau)$ is intractable, and we use the Posterior Trait Tracker $q_\phi(\mathbf{m}|\tau)$ to approximate it.

Here, we derive a lower bound for the log evidence $\log p_{\psi,\phi,\theta}(\tau)$:

$$
\begin{aligned}
\log p_{\psi,\phi,\theta}(\tau) &= \mathbb{E}_{q_\phi(\mathbf{m}|\tau)}[\log p_{\psi,\theta}(\tau)] \\
&= \mathbb{E}_{q_\phi(\mathbf{m}|\tau)}[\log \frac{p_{\psi,\theta}(\tau, \mathbf{m})}{p_{\psi,\theta}(\mathbf{m}|\tau)}] \\
&= \mathbb{E}_{q_\phi(\mathbf{m}|\tau)}[\log \frac{p_{\psi,\theta}(\tau, \mathbf{m})}{q_\phi(\mathbf{m}|\tau)}] + \mathbb{E}_{q_\phi(\mathbf{m}|\tau)}[\log \frac{q_\phi(\mathbf{m}|\tau)}{p_{\psi,\theta}(\mathbf{m}|\tau)}] \\
&= \mathbb{E}_{q_\phi(\mathbf{m}|\tau)}[\log \frac{p_{\psi,\theta}(\tau, \mathbf{m})}{q_\phi(\mathbf{m}|\tau)}] + D_{KL}(q_\phi(\mathbf{m}|\tau)||p_{\psi,\theta}(\mathbf{m}|\tau)) \\
&\geq \mathbb{E}_{q_\phi(\mathbf{m}|\tau)}[\log \frac{p_{\psi,\theta}(\tau, \mathbf{m})}{q_\phi(\mathbf{m}|\tau)}] \\
&= \mathbb{E}_{q_\phi(\mathbf{m}|\tau)}[\log p_{\psi,\theta}(\tau, \mathbf{m}) - \log q_\phi(\mathbf{m}|\tau)] \\
&= \mathbb{E}_{q_\phi(\mathbf{m}|\tau)}[\log p_{\psi,\theta}(\tau|\mathbf{m}) + \log p(\mathbf{m}) - \log q_\phi(\mathbf{m}|\tau)] \\
&= \mathbb{E}_{q_\phi(\mathbf{m}|\tau)}[\log p_{\psi,\theta}(\tau|\mathbf{m})] - \mathbb{E}_{q_\phi(\mathbf{m}|\tau)}[\log \frac{q_\phi(\mathbf{m}|\tau)}{p(\mathbf{m})}] \\
&= \mathbb{E}_{q_\phi(\mathbf{m}|\tau)}[\log p_{\psi,\theta}(\tau|\mathbf{m})] - D_{KL}(q_\phi(\mathbf{m}|\tau)||p(\mathbf{m})) \\
&= \log p(s_0^{\text{obs}}) + \underbrace{\sum_{t=0}^{T-1} \mathbb{E}_{q_\phi(\mathbf{m}|\tau)}[\log p_\psi(s_{t+1}^{\text{obs}}|s_t^{\text{obs}}, \pi_t, \mathbf{m})]}_{\text{State Prediction Likelihood}}
\end{aligned}
$$

$$
+ \underbrace{\sum_{t=0}^{T-1} \mathbb{E}_{q_\phi(\mathbf{m}|\tau)}[\log \beta_\theta(\pi_t|s_{\leq t}^{\text{obs}})]}_{\textit{principal}\text{ policy likelihood}} - \underbrace{D_{KL}(q_\phi(\mathbf{m}|\tau)||p(\mathbf{m}))}_{\text{Regularization Term}}.
$$

# B  Trait Inference Confusion Matrices

In our framework, a "trait" ($m$) represents a persistent, intrinsic characteristic of an agent. In tasks such as AdaSociety, it manifests as an agent's inherent production capability, while the SWM-AP method infers these unobservable attributes (e.g., skills, preferences, or risk attitudes) from agent interaction trajectories, shaping their behavior. This trait is not directly encapsulated in the observation space ($s_t$), rendering it inaccessible to the principal. We argue that the necessity of modeling such traits is directly tied to the degree of observational ambiguity in a system. In real-world scenarios, a principal (e.g., a government or platform) must make decisions under conditions of incomplete and ambiguous information. An individual's observable state ($s_t$) at any given moment, such as their current location or recent purchase, is often a highly ambiguous signal of their underlying trait ($m$), such as risk aversion, long-term preferences, or intrinsic skills. For instance, two individuals might be observed in the same location ($s_t$), but one is a risk-averse local resident (trait $m_1$) while the other is an adventurous tourist (trait $m_2$). Their immediate responses ($a_t$) and future state ($s_{t+1}$) to a new incentive (e.g., dynamic pricing) will diverge drastically, and predicting this divergence is impossible without inferring their underlying traits.

To test this hypothesis and explore trait interpretability under observational ambiguity, we conducted diagnostic experiments in the AdaSociety task. In standard training, the Posterior Trait Tracker produces a confusion matrix with clear diagonal dominance, indicating successful differentiation between agent types(Figure 6, left). However, non-trivial off-diagonal values suggest room for improvement, attributable to "modeling shortcuts". In later episode stages, the environment's predictability allows the model to rely on state history rather than precise trait inference, achieving low prediction error. To address this, we trained the tracker exclusively on high-ambiguity initial states, where agents' fixed starting positions provide minimal clues about their randomly assigned types. The resulting confusion matrix (Figure 6, right) exhibits stronger diagonal dominance. This demonstrates that SWM-AP's ability to learn interpretable traits is significantly enhanced under high-ambiguity conditions, which directly addresses persistent informational asymmetry in real-world scenarios.

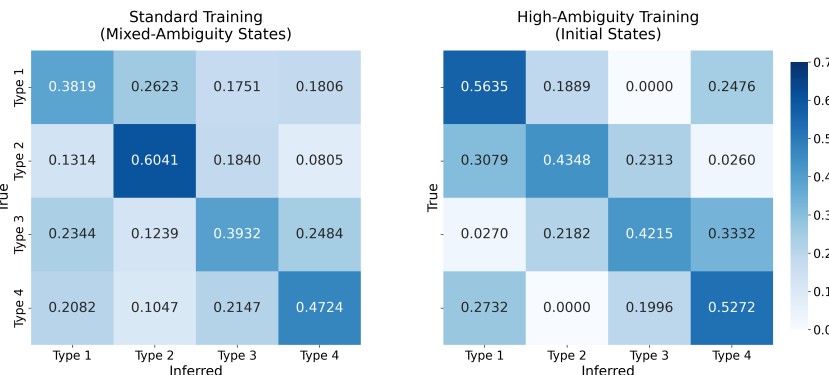

Figure 6: Confusion matrices for trait inference in AdaSociety. Left: Standard training on mixed-ambiguity states, showing moderate diagonal dominance due to modeling shortcuts in low-ambiguity late-episode states. Right: High-ambiguity training on initial states, demonstrating improved trait disentanglement with stronger diagonal dominance.

# C  Performance and Efficiency Results

To assess the scalability of SWM-AP in larger multi-agent settings, we extended the Facility Location task to 32 agents on a 7x7 grid with 5 placeable facilities. Each agent was randomly assigned one of two unobservable latent traits at the start of each episode, governing their behavior based on distance and congestion, with fixed home locations. The principal made 5 mechanism design decisions per episode. Results, averaged over 3 independent runs with different random seeds, are presented in Table 1. Efficiency is measured as the number of training steps required to reach MBPO's final performance (reward of 6.43), with lower values indicating better sample efficiency.

Table 1: Performance and efficiency results for the 32-agent Facility Location task.

| Algorithm | Final Reward (Mean $\pm$ Std) | Steps to MBPO Final Perf. |
|---|---|---|
| PPO | $6.55 \pm 0.03$ | 353,600 |
| MBPO | $6.43 \pm 0.04$ | 433,600 |
| Dreamer | $6.57 \pm 0.06$ | 300,800 |
| SWM-AP | $6.62 \pm 0.06$ | 274,667 |

These results demonstrate that SWM-AP achieves a superior sample efficiency compared to baselines.

# D  Computational Benchmarks

We report the runtime and memory usage benchmarked against the number of agents. The results indicate that both the runtime and memory footprint of our method fall within an acceptable range.

Table 2: Computational benchmarks for the Facility Location task across different agent numbers, per 100k training steps.

| Agents | Training Time (hrs) | Memory Footprint (MB) |
|---|---|---|
| 2 | 0.91 | 538 |
| 4 | 1.15 | 618 |
| 8 | 1.93 | 898 |
| 16 | 3.45 | 1786 |
| 32 | 5.25 | 5104 |

# E  Experimental Details

This appendix provides essential details for reproducibility. Key configurations and hyperparameters are summarized below. Table 3 summarizes crucial parameters for the experimental environments and core algorithms.

Table 3: Key Environment and Algorithm Configurations.

| Category | Parameter | Facility Location | Team Structure Optimization | Tax Adjustment |
|---|---|---|---|---|
| *Environment Specifics* | | | | |
| | Env. Source | Matrix | AdaSociety | AI Economist |
| | Agent Count | 8 | 4 | 4 |
| | Latent Trait Count. | 256 | 4 | 4 |
| | Mechanism Action | Select a point from Map(8*8) | Assign a team structures among 14 different types | Set a tax rate for each of the 7 tax brackets |
| | Episode Length | 5 | 50 | 1000 (for agents), 10 (for planner) |
| *SWM-AP: Social World Model (SWM)* | | | | |
| | Latent Inference Arch. | MLP (L:2, H:512) | MLP (L:2, H:512) + LSTM (L:1, H:512) + MLP (L:2, H:512) | MLP (L:2, H:512) + LSTM (L:1, H:512) + MLP (L:2, H:512) |
| | Dynamics Predict. Arch. | MLP (L:3, H:256) | GCN (L:3, H:[64, 128]) + MLP (L:2, H:128) | GCN (L:3, H:[64, 128]) + MLP (L:2, H:128) |
| | SWM Optimizer & LR | Adam, $10^{-3}$ | Adam, $10^{-3}$ | Adam, $10^{-3}$ |
| *SWM-AP: Mechanism Design Policy (PPO based)* | | | | |
| | Policy/Value Arch. | MLP (L:2, H:128) | GCN (L:3, H:[64, 64]) + MLP (L:2, H:256) | MLP (L:2, H:256) + LSTM(L:1,H:256) + MLP (L:1, H:256) |
| | Optimizer & LR | Adam, $2.5 \times 10^{-4}$ | Adam, $5 \times 10^{-4}$ | Adam, $1 \times 10^{-4}$ |
| | Discount ($\gamma$) | 0.99 | 0.99 | 0.99 |
| | Imagined Rollout (SWM) | 5 steps | 50 steps | 1000 steps |
| *Baselines: PPO* | | | | |
| | Policy/Value Arch. | MLP (L:2, H:128) | GCN (L:3, H:[64, 64]) + MLP (L:2, H:256) | MLP (L:2, H:256) + LSTM(L:1,H:256) + MLP (L:1, H:256) |
| | Optimizer & LR | Adam, $2.5 \times 10^{-4}$ | Adam, $5 \times 10^{-4}$ | Adam, $1 \times 10^{-4}$ |
| | Discount ($\gamma$) | 0.99 | 0.99 | 0.99 |
| *Baselines: MBPO* | | | | |
| | Policy/Value Arch. | MLP (L:2, H:128) | GCN (L:3, H:[64, 64]) + MLP (L:2, H:256) | GCN(L:3, H:[64, 64]) + MLP (L:2, H:256) |
| | Optimizer & LR | Adam, $2.5 \times 10^{-4}$ | Adam, $5 \times 10^{-4}$ | Adam, $5 \times 10^{-4}$ |
| | Discount ($\gamma$) | 0.99 | 0.99 | 0.99 |
| *General Training & Compute* | | | | |
| | Total Timesteps | $10^6$ | $1 \times 10^8$ | $5 \times 10^8$ |
| | Num. Random Seeds | 3 | 3 | 3 |
| | Error Bars | $\pm$ SEM over 3 runs | $\pm$ SEM over 3 runs | $\pm$ SEM over 3 runs |
| | GPUs Used | NVIDIA RTX 3090 (1 per run) | NVIDIA RTX 3090 (1 per run) | NVIDIA A100 (1 per run) |

**Further Details on SWM-AP:** SWM is trained to minimize Mean Squared Error for dynamics prediction, potentially with an additional VAE-like loss for trait inference if applicable. **Further Details on Baselines:** Model-free baselines like PPO implemented with standard configurations. For PPO, a clipping epsilon of 0.2 was used.

