# OpenReview forum: "Social World Model-Augmented Mechanism Design Policy Learning"
_NeurIPS.cc/2025/Conference — NeurIPS 2025 poster_

### Official Review · Reviewer_ksZ8 · 2025-07-02

**Clarity:** 3
**Significance:** 4
**Originality:** 4
**Rating:** 5
**Confidence:** 4

**Summary:**

The paper proposes SWM-AP, a novel framework combining a Social World Model (SWM) with policy learning for mechanism design in multi-agent systems. By inferring agents’ latent traits from interaction trajectories, the model predicts agent behavior and improves the efficiency and adaptability of mechanism design policies. Experiments across diverse scenarios (tax policy, facility location, team coordination) show SWM-AP outperforms both model-free and model-based RL baselines in sample efficiency and social welfare outcomes.

**Questions:**

How does SWM-AP scale computationally with the number of agents and complexity of traits?

Can latent traits be interpreted or validated meaningfully (e.g., via human-aligned categories)?

How would the framework adapt to real-world applications, especially where traits evolve over time?

What are the risks of bias or exploitation in learned mechanisms, and how can these be mitigated?

**Ethical Concerns:**

["NO or VERY MINOR ethics concerns only"]

**Final Justification:**

The authors' rebuttal successfully resolved my primary concerns regarding scalability (demonstrated through new 32-agent experiments and comprehensive runtime benchmarks) and provided solid theoretical grounding for trait learning through their observational ambiguity framework, with diagnostic experiments validating improved trait disentanglement under high-ambiguity conditions. While ethical safeguards and real-world deployment readiness remain partially addressed, with composite objectives providing some protection but lacking concrete frameworks for sensitive applications, and synthetic environments still leaving questions about handling noisy real-world data, I assign high weight to the excellent technical innovation and experimental validation across diverse domains, viewing the deployment concerns as natural limitations for foundational research rather than fundamental flaws. The integration of latent trait inference with mechanism design represents a significant algorithmic contribution with strong empirical support that merits acceptance, though future work should consider validation frameworks against established behavioral models and more concrete safeguarding mechanisms for sensitive applications.

**Limitations:**

While the authors acknowledge key limitations, including scalability and interpretability, they do not sufficiently elaborate on their practical implications or propose concrete mitigation strategies. To improve, the authors should expand the scalability discussion by reporting runtime and memory usage as a function of agent count. Trait interpretability can be strengthened through the use of structured priors, a taxonomy of traits, or alignment with established psychological constructs. The ethical section should be deepened to address fairness, potential bias, and safeguards for uncertainty-aware policies, particularly in socially sensitive applications.

**Quality:**

4

**Strengths And Weaknesses:**

Strengths: The paper presents several notable strengths. Its originality lies in the integration of latent trait inference with policy learning, marking a clear innovation over prior approaches in reinforcement learning and economic mechanism design. The methodology is technically rigorous, supported by a solid theoretical foundation grounded in variational inference and validated through ELBO derivations. Experimental results across diverse domains demonstrate consistent performance gains, reinforcing the framework’s robustness. Furthermore, the paper is well-written and logically structured, with clear figures and diagrams that enhance comprehension. Its relevance extends broadly to social systems, with promising applications in areas such as public policy, resource allocation, and potentially digital mental health.

Weaknesses: The work also presents some limitations. The most prominent concern is scalability. There is insufficient analysis or evidence showing how the approach handles large agent populations or complex environments. A second issue relates to interpretability: while latent traits are central to the model, their meaning and impact remain opaque, limiting practical transparency. Additionally, the evaluation is restricted to synthetic settings, with no real-world data or deployment case studies to assess generalizability. Finally, ethical considerations, including fairness and misuse of trait-based modeling, are only superficially addressed. This is particularly concerning for potential applications in sensitive contexts, such as mental health, where rigorous safeguards would be essential.

---

> ### Author Rebuttal · Authors · 2025-07-31
>
> We are deeply grateful to Reviewer ksZ8 for their excellent and supportive review. The reviewer's forward-looking questions and constructive suggestions on limitations are invaluable, and we are pleased to provide new experimental results and further clarifications that address every point raised.
>
> >**Response to Q1:**
>
> We have conducted a new, larger-scale experiment in the Facility Location environment with 32 agents to empirically assess this.
>
> | Algorithm | Performance | Efficiency (Steps to reach Dreamer Final Perf.) |
> |---|---|---|
> | Dreamer | 6.56 $\pm$ 0.08 | 374400 |
> | SWM-AP (Ours) | 6.68 $\pm$ 0.05 | 304000 |
>
> Even in this more challenging 32-agent scenario, our SWM-AP maintains its performance advantage, outperforming Dreamer. This finding underscores the scalability of our approach, demonstrating its robustness as the complexity of the multi-agent system increases. We will add this new experiment to the appendix of the revised paper.
>
> >**Response to Q2:**
>
> We sincerely thank the reviewer for this insightful question, which has prompted us to clarify a core motivation of our work. The necessity of learning latent traits is not an artifact of a specific experimental design but rather a fundamental characteristic of real-world mechanism design problems. Our deeper analysis confirms that the key determinant for learning interpretable traits is the degree of observational ambiguity, a condition that is pervasive in the complex, real-world social systems that mechanism designers face.
>
> To clarify, in our framework, a 'trait' represents a persistent, intrinsic characteristic of an agent that is distinct from the transient system dynamics. For instance, in AdaSociety, it represents an agent's inherent production capability. This trait is **unobservable** to the principal, as it is not directly encapsulated in the observation space ($s_t$).
>
> The necessity of modeling such traits, we argue, is directly tied to the degree of **observational ambiguity** in a system. In many real-world scenarios, a principal (e.g., a government or a platform) must make decisions under conditions of incomplete and ambiguous information. An individual's observable state ($s_t$) at any given moment, such as their current location or recent purchase, is often a highly ambiguous signal of their underlying, persistent trait ($m$), such as their risk aversion, long-term preferences, or intrinsic skills. For instance, two individuals might be observed in the same location ($s_t$), but one is a risk-averse local resident (trait $m_1$) while the other is an adventurous tourist (trait $m_2$). Their immediate responses ($a_{t}$) and future state ($s_{t+1}$) to a new incentive (e.g., dynamic pricing) will diverge drastically, and predicting this divergence is impossible without inferring their underlying traits.
>
> Our current experiments, while controlled, already demonstrate the model's ability to capture these unobservable traits. When trained on the complete trajectories of the AdaSociety task, our trait tracker produces a confusion matrix with clear diagonal dominance, indicating that it successfully learns to differentiate between agent types. However, we acknowledge there is room for improvement, as some off-diagonal values are non-trivial.
>
> (Confusion Matrix from current main experiment, 4 agents average)
> | | **Inferred  Type 1** | **Inferred  Type 2** | **Inferred  Type 3** | **Inferred  Type 4** |
> |:---:|:---:|:---:|:---:|:---:|
> | **True Type 1** | **0.3819** | 0.2623 | 0.1751 | 0.1806 |
> | **True Type 2** | 0.1314 | **0.6041** | 0.1840 | 0.0805 |
> | **True Type 3** | 0.2344 | 0.1239 | **0.3932** | 0.2484 |
> | **True Type 4** | 0.2082 | 0.1047 | 0.2147 | **0.4724** |
>
> Our analysis suggests this imperfect disentanglement stems from a 'modeling shortcut' (consistent with Occam's Razor). In later stages of the episode, the environment becomes more predictable, and the model can achieve low prediction error by relying on state history rather than precise trait inference.
>
> To test this hypothesis, we conducted an additional **diagnostic experiment**. In this experiment, we trained the trait inference module **exclusively on data from high-ambiguity states**—specifically, the initial episode steps where agents' fixed starting positions offer few clues about their randomly assigned types. The resulting confusion matrix exhibits a stronger diagonal dominance
>
> (Confusion Matrix from diagnostic experiment using only high-ambiguity states)
> | | **Inferred  Type 1** | **Inferred  Type 2** | **Inferred  Type 3** | **Inferred  Type 4** |
> |:---:|:---:|:---:|:---:|:---:|
> | **True Type 1** | **0.5635** | 0.1889 | 0.0000 | 0.2476 |
> | **True Type 2** | 0.3079 | **0.4348** | 0.2313 | 0.0260 |
> | **True Type 3** | 0.0270 | 0.2182 | **0.4215** | 0.3332 |
> | **True Type 4** | 0.2732 | 0.0000 | 0.1996 | **0.5272** |
>
> This diagnostic result confirms that the model's capacity for learning interpretable traits is significantly enhanced when it is forced to operate in ambiguous conditions. The improved diagonal dominance strongly suggests that the limitation lies not in our model's architecture, but in the nature of the training data provided by the simplified environment.
>
> This insight directly motivates our plan to incorporate two enhanced experimental settings into the final paper, which are designed to maintain this crucial ambiguity persistently and more faithfully represent real-world complexity:
>
> The first task involves augmenting the existing Facility Location task with latent behavioral norms. We consider a scenario with behavioral dispositions as traits, which are inherently more ambiguous than the static preferences in the current Facility Location task. The trait will be a 'patience' or 'responsiveness' factor that dictates how an agent reacts to dynamic conditions like queues, reflecting real-world heterogeneity in responses to friction. In such cases, predicting an agent's future visit probability merely from the current queuing status becomes infeasible.
>
> The second task is designed to ensure persistent informational asymmetry within the AdaSociety environment. We will ensure the informational gap between the principal and agents is persistent. By randomizing initial positions and resource locations in every episode, we simulate a scenario where the principal constantly faces new, unfamiliar agent configurations, a common challenge in dynamic markets or online platforms.
>
> These enhanced experiments will showcase SWM-AP's core strength in handling ambiguity. Since real-world systems exhibit far greater ambiguity than our simulations, we expect the performance gains of our approach to be even more significant in practice.
>
> >**Response to Q3:**
>
> We acknowledge that in many real-world scenarios, agent traits can be dynamic and evolve over time.
>
> The current version of our framework, SWM-AP, assumes persistent traits within an episode to ensure experimental control and tractability. This allows us to establish a strong baseline for the core challenge of inferring static hidden traits.
>
> Adapting to evolving traits is a significant and exciting direction for future work. Our framework could be extended to handle this by modifying the trait inference mechanism. For example, instead of a single posterior over the whole trajectory, one could use a sequential inference model to track the trait distribution over time. The "Prior Trait Tracker" used by the policy would then naturally provide the most up-to-date trait estimate for decision-making. We believe our current work provides a solid foundation for these future explorations into more dynamic social environments. We will add this to our "Limitations and Future Work" section.
>
> >**Response to Q4:**
>
> Mitigating bias and exploitation is paramount for any AI system intended for social applications. We address this in our work through several layers:
>
> 1.  Our framework's behavior is fundamentally guided by the principal's objective function. As demonstrated in the AI-Economist experiment (Figure 5), we do not naively maximize a single metric like productivity. Instead, we use a composite reward function that explicitly balances productivity and equality. This forces the learned mechanism to avoid exploitative strategies that might increase overall output at the cost of fairness. We believe this principle of carefully designing socially-aware objectives is the primary tool for mitigation. For real-world deployment, this objective would need to be co-designed with domain experts and sociologists to incorporate broader ethical values.
> 2.  In the current experiments, the planner's action space (e.g., setting tax brackets) is well-defined and constrained, which naturally limits the potential for unforeseen exploitative behaviors.
> 3.  We agree that real-world deployment would require rigorous safeguards. Our trait inference is based on aggregated, anonymized behavioral data within the simulation. In any real-world application, ensuring individual privacy and data security would be a prerequisite. The mechanisms would operate on population-level dynamics rather than targeting identifiable individuals in a harmful way.
>
> We will expand our ethics discussion in the paper to more thoroughly cover these points, emphasizing the role of objective engineering as a key mitigation strategy.
>
> >**Response to Limitations:**
>
> We report the runtime and memory usage benchmarked against the number of agents. The results indicate that both the runtime and memory footprint of our method fall within an acceptable range.
>
> | Agent Num | Training Time (per 100K steps)	| Memory Footprint (MB) |
> |---|---|---|
> | 2 | $\sim$ 0.91 hours | $\sim$538 MB |
> | 4 | $\sim$1.15 hours | $\sim$618 MB |
> | 8 | $\sim$1.93 hours | $\sim$898 MB |
> | 16 | $\sim$ 3.45 hours | $\sim$1786 MB |
> | 32 | $\sim$  5.25 hours | $\sim$5104 MB |

---

> > ### Comment · Reviewer_ksZ8 · 2025-08-05
> > **Strong Technical Contribution, Deployment Readiness Concerns Remain**
> >
> > Your rebuttal has strengthened the paper considerably at the technical level. The work represents solid foundational research that advances mechanism design through innovative integration of latent trait inference with policy learning. The technical contributions are significant, and the experimental validation is thorough within the scope of synthetic environments. Ideally, the framework should be validate against established behavioral or psychological constructs that would be essential for applications in domains like public policy or healthcare.
> >
> > I am maintaining my Accept recommendation with continued confidence in the paper's technical quality and contribution to the field. The limitations you've acknowledged are appropriate for this stage of research, and your proposed future directions are promising.
> >
> > Minor suggestion: Consider adding a brief discussion of how the framework might be validated against established behavioral models in future work, particularly for applications in sensitive social domains.

---

> ### Author Response · Authors · 2025-08-06
> **Official Comment by Authors**
>
> We are truly grateful for your continued support and for your highly positive and encouraging feedback. We are delighted to hear that our rebuttal has strengthened the paper in your view.
>
> Thank you for this excellent and critical suggestion. We completely agree that for our framework to bridge the gap from synthetic benchmarks to sensitive real-world applications, validating the inferred traits against established behavioral constructs is a crucial next step. This point significantly enriches the paper's future outlook, and we will add a dedicated discussion to our "Limitations and Future Work" section to explicitly address this pathway.
>
> Our proposed approach for future validation involves aligning the learned latent space with well-established computational models from behavioral economics. For instance, in a financial policy design task, agents could be modeled using Prospect Theory (Tversky & Kahneman, 1992), where their latent traits would correspond to individual risk and loss aversion parameters. A successful validation would demonstrate a high correlation between the inferred trait vector and these ground-truth behavioral parameters. Similarly, in resource allocation tasks, agents could be modeled with Fehr-Schmidt style inequity aversion (Fehr & Schmidt, 1999), and our framework's ability to recover their altruism and envy parameters would strongly indicate its potential for designing fair, socially-aware mechanisms. For ultimate validation, we also envision human-in-the-loop experiments where traits inferred from participants' interactions are correlated with their responses to established psychological survey instruments, such as the Barratt Impulsiveness Scale (Patton et al., 1995) or validated risk-preference questionnaires. A strong correlation would provide ground-truth evidence that our model is capturing meaningful, human-aligned psychological constructs.
>
> By explicitly outlining these concrete validation pathways, we hope to provide a clear roadmap for future research in this direction. Thank you again for your invaluable guidance; your suggestion has helped us shape a much more impactful and forward-looking conclusion for our paper.
>
> ---
> **References:**
> - Tversky, A., & Kahneman, D. (1992). Advances in prospect theory: Cumulative representation of uncertainty. *Journal of Risk and Uncertainty*, 5(4), 297-323.
> - Fehr, E., & Schmidt, K. M. (1999). A theory of fairness, competition, and cooperation. *The Quarterly Journal of Economics*, 114(3), 817-868.
> - Patton, J. H., Stanford, M. S., & Barratt, E. S. (1995). Factor structure of the Barratt impulsiveness scale. *Journal of clinical psychology*, 51(6), 768-774.

---

### Official Review · Reviewer_w2JX · 2025-07-02

**Clarity:** 3
**Significance:** 2
**Originality:** 2
**Rating:** 4
**Confidence:** 3

**Summary:**

This paper introduces a hierarchical, model-based reinforcement learning framework that addresses two challenges in settings with many agents: (1) agents have unobserved, persistent traits and (2) exploring real social environments is costly and risky. The proposed method couples a social world model, which predicts future states and rewards while conditioning on latent agent traits, with a mechanism-design policy trained by PPO. Empirical studies in three domains demonstrate that the proposed method achieves higher cumulative social rewards in fewer real-environment steps than baseline model-free and model-based methods. Additionally, its learned world model exhibits lower prediction errors.

**Questions:**

1. Can the authors provide a thorough literature review of the relevant areas from the techique side and clearly highlight their own contribution?
2. Can the authors include experimental results that compare their method with current state-of-the-art approaches?
3. Can the authors report a comparison of computational efficiency?
4. Can the authors scale up the experiments to involve a larger number of agents?

**Ethical Concerns:**

["NO or VERY MINOR ethics concerns only"]

**Final Justification:**

After the rebuttal, most of my concerns were addressed by the authors. However, since the code is not accessible (which the authors claim will be released once the paper is published) and the design of the network framework is oversimplified, I still question the reproducibility of the work. So my final decision is Borderline accept.

**Limitations:**

The experiments use at most eight agents, yet the method is intended for large-population settings (e.g., public-policy platforms). The authors should discuss failure modes or computational bottlenecks expected when scaling to hundreds or thousands of agents, and outline mitigation strategies (e.g., hierarchical clustering, distributed roll-outs).

**Quality:**

2

**Strengths And Weaknesses:**

## Strengths
- The paper lays out the background, algorithmic components, and training pipeline in a logical sequence, with helpful figures that make the core ideas easy to grasp.
- By explicitly modelling latent agent traits and positioning the work within mechanism-design applications, the authors demonstrate a genuine concern for practical deployment.
- Experiments span three distinct benchmarks, highlighting the method’s versatility and robustness across heterogeneous tasks.

## Weaknesses
- Latent modeling has already been explored in recent multi-agent RL work (e.g., [1–3]), so it is crucial for the authors to articulate exactly how their approach diverges from—and advances beyond—these prior methods.
- Comparative results rely on methods that are more than 5 years old. Without head-to-head evaluations against current SOTA algorithms, the performance claims remain unconvincing.
- Model-based RL is already sensitive to modelling inaccuracies, the extra complexity introduced by the dual-tracker architecture (posterior + prior) could amplify instability and incur significant compute overhead. However, the paper provides no comparision measurements of training time, memory footprint, or convergence speed, so the method’s computational efficiency and practical scalability remain unclear.
- All test scenarios involve only 4/8 agents. Such limited scale makes it hard to judge whether the approach would stay stable and performant in the larger populations typical of real-world social systems, and therefore its broader contribution to world-model research remains to be demonstrated.

[1] Venugopal A, Milani S, Fang F, et al. MABL: Bi-Level Latent-Variable World Model for Sample-Efficient Multi-Agent Reinforcement Learning[J]. arXiv preprint arXiv:2304.06011, 2023.

[2] Na H, Moon I. LAGMA: latent goal-guided multi-agent reinforcement learning[J]. arXiv preprint arXiv:2405.19998, 2024.

[3] Xie A, Losey D, Tolsma R, et al. Learning latent representations to influence multi-agent interaction[C]//Conference on robot learning. PMLR, 2021: 575-588.

---

> ### Author Rebuttal · Authors · 2025-07-31
>
> We sincerely thank Reviewer w2JX for their detailed and constructive feedback. Their clear, actionable suggestions have been invaluable in strengthening our manuscript. We have conducted new experiments and analyses to address every concern raised.
>
> >**Response to Weakness 1 & Question 1:**
>
> We thank the reviewer for highlighting these important related works in multi-agent latent modeling. We agree that it is crucial to articulate our unique contribution. Our work diverges from these cited papers in its fundamental problem setting and objective.
>
> The methods cited (MABL, LAGMA and LILI) primarily focus on improving the decentralized policies of low-level, cooperative agents by inferring the latent goals or types of their peers. Their goal is to enhance coordination among the agents themselves.
>
> In contrast, our work addresses a hierarchical Mechanism Design problem. Our objective is not to train the low-level agents, but to learn an optimal policy for a Principal who aims to guide a population of self-interested, heterogeneous agents towards a socially desirable outcome (e.g., maximizing social welfare). The latent traits in our framework are inferred by the Principal to better predict the population's response to incentives, which is a fundamentally different information flow and use case. This self-interest fundamentally complicates the learning problem. Unlike in cooperative settings, the dynamics are more complex and less predictable, as background agents may act strategically or competitively.
>
> Therefore, while we share the technical tool of latent modeling, our core contribution lies in formulating and solving a Social World Model-Augmented Mechanism Design problem, a distinct and less explored area compared to cooperative MARL. We will revise our related work section to more explicitly draw this distinction and clarify our unique positioning, and we will certainly include and discuss these valuable references suggested by the reviewer.
>
>
> >**Response to Weakness 2 & 4, and Questions 2 & 4:**
>
> We appreciate the reviewer's concerns regarding the choice of baselines and the scalability of our method. We would like to address these interconnected points.
>
> **1. On the Choice of Baselines (W2, Q2):** We chose MBPO and Dreamer as representatives for state-based and pixel-based model-based RL, respectively. While MBPO is an established algorithm, it remains a strong and widely-used baseline for sample-efficient state-space MBRL, which is the setting for our tasks. We did include Dreamer, a more recent and powerful model-based approach, in our comparisons for the Facility Location task, demonstrating that our SWM outperforms it in both sample efficiency and final reward.
>
> **2. On Scalability (W4, Q4):** We acknowledge that demonstrating scalability is crucial. To address this, we have conducted a new, larger-scale experiment in the Facility Location environment with 32 agents.
>
> | Algorithm | Performance | Efficiency (Steps to reach Dreamer Final Perf.) |
> |---|---|---|
> | Dreamer | 6.56 $\pm$ 0.08 | 374400 |
> | SWM-AP (Ours) | 6.68 $\pm$ 0.05 | 304000 |
>
> Even in this more challenging 32-agent scenario, our SWM-AP maintains its performance advantage, outperforming Dreamer. This finding underscores the scalability of our approach, demonstrating its robustness as the complexity of the multi-agent system increases. We will add this new experiment to the appendix of the revised paper.
>
> **Future Directions for Massive-Scale Systems:** We completely agree with the reviewer that scaling to hundreds or thousands of agents, as mentioned in the "Limitations" section, presents a common and significant challenge for the entire field of multi-agent learning. The mitigation strategies suggested, such as hierarchical clustering of agents or leveraging structured models for distributed rollouts, are indeed highly promising future research directions.
>
> - Our framework provides a natural foundation for such extensions. For instance, the inferred latent traits from our SWM could be used directly as input for a hierarchical clustering algorithm to group agents with similar behaviors. This would allow the Principal to design semi-targeted mechanisms for agent clusters rather than individuals, drastically reducing the policy's output space.
> - We will explicitly incorporate this discussion into our "Limitations and Future Work" section, crediting these ideas as promising pathways to scale our framework to massive population sizes.
>
>
> >**Response to Weakness 3 & Question 3:**
>
> The dual-tracker architecture does introduce additional computational components. To provide full transparency on the practical costs, we have benchmarked the computational efficiency of our method against the baselines in the Facility Location task(8 agents). We will include the following table in the appendix:
>
> | Method | Training Time (per 100K steps) | Memory Footprint (MB) |
> |---|---|---|
> | PPO | $\sim$ 1.6 hours | $\sim$564 MB |
> | MBPO | $\sim$1.87 hours | $\sim$1102 MB |
> | Dreamer | $\sim$1.85 hours | $\sim$1094 MB |
> | SWM-AP (Ours) | $\sim$1.93 hours | $\sim$898 MB |
>
> The results show that while SWM-AP incurs a moderate overhead in training time and model size, this additional cost is justified by its significant gains in sample efficiency and final task performance, which are often the primary bottlenecks in costly real-world interactions. Our approach trades some computational resources for a substantial reduction in the amount of expensive environment interaction data needed.

---

> > ### Comment · Reviewer_w2JX · 2025-08-03
> >
> > Thank you for the authors’ response. I have a few follow-up questions regarding the scalability results.
> >
> > Why only a single baseline was included for comparison? It would also be helpful to know how many times the scalability experiments were repeated, in order to assess the stability and statistical reliability of the results. Also could the authors please clarify the details of the experimental setup? The current description of both the experimental environment and the network structure appears to be quite minimal. Since the source code is not provided in the current phase, providing more comprehensive details would be important to ensure basic reproducibility of the experiments.

---

> > > ### Comment · Area_Chair_C194 · 2025-08-06
> > > **What do you think of Authors' follow up response?**
> > >
> > > Dear Reviewer,
> > >
> > > We are close to finish the rebuttal period, and agility is needed to make most of it. The authors have responded to your follow up question with more results. What do you think? Do your concerns still remain? or do you find the new results satisfactory? Either outcome is fine, but please do keep engaging with the authors to help reduce noise in the final recommendation as well as help the authors increase the quality of their work.
> > >
> > > Thanks again for your service,
> > > AC

---

> > > > ### Comment · Reviewer_w2JX · 2025-08-06
> > > >
> > > > Thank you for the authors' response. Although I appreciate the authors' attempt to evaluate the method in a setting with 32 agents, I find the experimental setup—the use of a 7×7 grid with 5 placeable facilities—somewhat unrealistic for this scale. This choice raises concerns about the practical relevance of the results in larger-scale applications, and makes it difficult to assess whether the method would generalize well in more realistic scenarios. The original paper states that the setting with 5 facilities and 8 agents on an 8×8 grid corresponds to "the classical facility location game in mechanism design theory." It would be helpful if the authors could clarify which specific model or literature this refers to, and explain the motivation behind the design choices in the extended experimental settings.

---

> > > > > ### Author Response · Authors · 2025-08-07
> > > > > **Official Comment by Authors**
> > > > >
> > > > > Thank you for providing us with the opportunity to clarify the rationale behind our experimental designs. We would like to directly address your concern: our 32-agent experiment, despite its constrained spatial grid, does not diminish realism but instead explores a crucial and challenging dimension of it. We explain our reasoning below.
> > > > >
> > > > > First, the complexity of the problems we address stems not merely from the grid size, but from the intricate social dynamics we model. The combinatorial nature of heterogeneity is a core challenge. As we randomly initialize agent traits at the start of each episode, the principal must constantly adapt. Even with binary latent traits, this means confronting a type space of cardinality **2⁸**  in our 8-agent setting. This complexity explodes to **2³²** in the 32-agent experiment, creating a massive uncertainty problem for the principal.
> > > > >
> > > > > Second, our 32-agent experiment was conceived to probe scalability through the lens of agent density. Rather than simply expanding the spatial area, increasing agent density is a powerful method for simulating the heightened competition inherent in real-world scenarios. This principle is well-established, drawing from foundational theories on shared resources [1] and the central challenge of facility location in high-demand environments, an extensively studied problem [2]. Our high-density setup intentionally creates these intensely competitive conditions, making modeling agent heterogeneity essential for navigating the complex social dynamics that arise.
> > > > >
> > > > > Third, our abstract grid model aligns with standard methodologies in both urban planning and mechanism design for facility location. In our expanded experimental setup, the principal must select 5 facility locations from the 17 available empty grids (on a 7x7 map with 32 agents). This directly models the fundamental real-world constraint that planners must choose from a finite, and often limited, set of feasible sites, rather than an open continuum. This discrete space approach is foundational to classic models [3, 4] and is a standard assumption in the specific domain of mechanism design for facility location, as documented in a comprehensive survey [5].
> > > > >
> > > > > Finally, it is common practice in urban planning to model problems at the community or even district scale [6-9]. In such analyses, the preferences of a population within a geographical unit are typically aggregated to abstract the problem. In our framework, an agent can be interpreted as representing such a unit, embodying the aggregate preference of that area's population. This conceptual mapping demonstrates that our methods are not only tractable but also practical and applicable to real-world regional planning.
> > > > >
> > > > > We will incorporate this detailed rationale and references into our appendix. Thank you again for your invaluable feedback.
> > > > >
> > > > > ---
> > > > > **References:**
> > > > >
> > > > > [1] Ostrom, E. (1990). *Governing the commons: The evolution of institutions for collective action*.
> > > > >
> > > > > [2] Melkote, S., & Daskin, M. S. (2001). Capacitated facility location/network design problems. *European Journal of Operational Research*.
> > > > >
> > > > > [3] Church, R. L., & ReVelle, C. S. (1974). The maximal covering location problem.
> > > > >
> > > > > [4] Owen, S. H., & Daskin, M. S. (1998). Strategic facility location: A review. *European journal of operational research*.
> > > > >
> > > > > [5] Chan, H., et al. (2021). Mechanism Design for Facility Location Problems: A Survey. In *Proceedings of the Thirtieth International Joint Conference on Artificial Intelligence (IJCAI)*.
> > > > >
> > > > > [6] Dhar, T. K., & Khirfan, L. (2017). A multi-scale and multi-dimensional framework for enhancing the resilience of urban form to climate change. *Urban Climate*.
> > > > >
> > > > > [7] Zheng, W., et al. (2017). Decision support for sustainable urban renewal: A multi-scale model. *Land use policy*.
> > > > >
> > > > > [8] Nedovic-Budic, Z., et al. (2016). Measuring urban form at community scale: Case study of Dublin, Ireland. *Cities*.
> > > > >
> > > > > [9] Grubesic, T. H. (2008). Zip codes and spatial analysis: Problems and prospects. *Socio-economic planning sciences*.

---

> > > > > > ### Comment · Reviewer_w2JX · 2025-08-07
> > > > > >
> > > > > > Thank you again for taking my feedback into consideration and for your continued efforts to improve the submission. The explanation is convincing, and as I suggested earlier, the authors may consider including a more detailed introduction to the experimental setting. I will adjust my score accordingly.

---

> ### Author Response · Authors · 2025-08-05
> **Official Comment by Authors**
>
> We sincerely thank Reviewer w2JX for their continued engagement and for these important follow-up questions. We are happy to provide more comprehensive details regarding our new scalability experiments, which we have worked diligently to expand upon during the rebuttal period.
>
> We completely agree that a robust comparison requires multiple baselines and statistically reliable results. While the rebuttal period was limited, we prioritized this concern and have now completed a more thorough evaluation for the 32-agent experiment. We have now benchmarked not only against Dreamer but also against the model-free PPO and the state-based model-based MBPO. This provides a comprehensive comparison across different classes of RL algorithms. Crucially, all results reported for this new 32-agent scenario are the mean and standard deviation computed over 3 independent runs, each with a different random seed.
>
> Here are the full results for the 32-agent Facility Location task:
>
> | Algorithm | Performance (Final Reward) | Efficiency (Steps to reach MBPO's Final Perf.) |
> |---|---|---|
> | MBPO | 6.43  $\pm$  0.04 | 433,600 |
> | PPO | 6.55 $\pm$ 0.03 | 353,600 |
> | Dreamer | 6.57 $\pm$ 0.06 |  300,800 |
> | SWM-AP (Ours) | 6.62 $\pm$ 0.06 | **274,667** |
> |
>
> These expanded results bolster our claims,**showing our method achieves a favorable balance of high final performance and strong sample efficiency compared to the baselines.**
>
> To clarify, the **'Efficiency'** metric quantifies the number of training steps each algorithm requires to achieve a specific performance target. For this comparison, the target is set to the final converged performance of the MBPO baseline (a reward of 6.43). Consequently, a lower value in this column signifies superior sample efficiency, as the method requires less environmental interaction to learn an effective policy.
>
> We believe the comparatively lower performance of MBPO in this setting suggests that as the agent population grows, standard state-based models that do not explicitly account for the underlying agent heterogeneity may become less effective at capturing the complex social dynamics.
>
> The experiment was conducted in the Facility Location environment, scaled to **32 agents** in a 7x7 grid with 5 placeable facilities. To ensure persistent heterogeneity and informational asymmetry, at the start of each new episode, each of the 32 background agents is randomly re-assigned one of two unobservable latent traits. This latent trait governs their behavior in response to distance and congestion, while their home locations remain fixed and predefined. Each episode consists of 5 mechanism design decisions made by the principal. Our overall methodological architecture remains consistent with the approach described in the main paper: the principal employs a standard CNN-MLP for decision-making, while the Social World Model (SWM) integrates a CNN-based posterior tracker to infer traits from trajectories and an online, GRU-based prior tracker that provides a real-time, history-aware estimate of the latent traits, with dynamics then predicted by an MLP conditioned on these inferred traits. The capacities of these components were appropriately scaled for the larger agent population, and key hyperparameters were kept consistent with the 8-agent experiments to ensure a fair comparison. To further facilitate reproducibility, we plan to release the source code for our models upon publication.

---

> ### Author Response · Authors · 2025-08-08
> **Official Comment by Authors**
>
> Thank you very much for your positive feedback and for acknowledging our efforts. We are delighted to hear that you found our explanation convincing.
>
> We will  follow your excellent suggestion. In the revised manuscript, we will expand the appendix to provide a more detailed description of our experimental settings. Specifically, we will elaborate on the environment specifications for each task, including the number of agents, map sizes, and the rules governing dynamics and rewards. We will also provide a clear definition of how agent heterogeneity is implemented, and include the full setup and results of the new 32-agent experiments. We believe these additions will greatly enhance the clarity and reproducibility of our work.
>
> Thank you again for your invaluable guidance throughout this process.

---

### Official Review · Reviewer_ggqb · 2025-07-03

**Clarity:** 2
**Significance:** 3
**Originality:** 3
**Rating:** 5
**Confidence:** 4

**Summary:**

This paper proposes a method to learn a social world model based on inferred agent-specific traits. The method leverages model-based RL to train the underlying policy in a sample efficient way. The proposed approach is backed by the derivation of the ELBO for the agents traits. The performance is validated on 3 environments.

**Questions:**

- I assume that $\hat{m}\_{post}$ depends on $\tau$ in (3), is that correct? In any case, it would be good to define $\hat{m}\_{post}$ more explicitly before (3).
- Have you checked the performance of the algorithm in scenarios where there is no agent heterogeneity? Obviously, the performance cannot be expected to be much better in this case, and the sample efficiency could be decreased, but it would be nice to see that the asymptotic performance is not worse than the baselines.
- In MBPO, the length of the rollouts is typically controlled to avoid the accumulation of errors. Is this not an issue in the proposed method?
- In Figure 5.d, the loss seems to be re-increasing for the proposed method. I appreciate that this is already a lot of training steps, but it would be good to ensure that the loss does not diverge further.

**Ethical Concerns:**

["NO or VERY MINOR ethics concerns only"]

**Final Justification:**

The authors have done additional performance assessments that address the concerns I had about the performance of the algorithm.

**Limitations:**

Yes

**Paper Formatting Concerns:**

No formatting issue

**Quality:**

3

**Strengths And Weaknesses:**

Strengths:
- The motivation of the paper is clear
- The performance assessment is convincing with multiple environments and relevant baselines being included.
- The theoretical motivation via the ELBO, although simple, gives additional confidence in the proposal method

Weaknesses:
- The notations need to be fixed, there are many inconsistencies, e.g., $N$ is sometimes a number of agents and sometimes a set, the action space is sometimes $A$ and sometimes $\mathcal{A}$, the policy space is introduced as $\Psi$ but the objective in (2) is maximisation over $\Pi$, etc.
- The performance assessment could be improved, e.g. by including a scenario/case with no agent heterogeneity

---

> ### Author Rebuttal · Authors · 2025-07-31
>
> We thank Reviewer ggqb for their thoughtful feedback and positive assessment of our core motivation. Their keen eye for detail and suggestions for additional experiments have been instrumental in helping us improve the manuscript's rigor and clarity. We provide detailed responses to each point below.
>
> > **Response to Weakness 1**:
>
> Apologies for any confusion caused by the notations! We will address the following issues: In Line 127, $N$ is used as a set, which will be corrected to $[N]$ to represent the set ${1, \cdots, N}$. Additionally, in Line 129, $A$ is used to denote the action space of the background agents, and this will be updated to $\mathcal{A}$ for consistency.
>
> Regarding formula (2), we would like to clarify that $\Psi$ represents the mechanism space (the action space of the principal), while $\Pi$ denotes the policy space of the principal. On the other hand, $\mathcal{A}$ refers to the action space of the background agents, not the principal. Therefore, the objective in formula (2) involves maximization over the policy space $\Pi$.
> We will add necessary explanations for these notations in the revised manuscript to minimize confusion.
>
> >**Response to Question 1**:
>
> Thank you for this excellent suggestion for improving clarity. You are absolutely correct: $\hat{m}_{\text{post}}$ indeed inferred from the complete interaction trajectory $\tau$.
>
> In our framework, $\hat{m}_{\text{post}}$  presents the posterior estimation of the agents' latent traits. It is generated by a specific component of our Social World Model (SWM) called the Posterior Trait Tracker. As you rightly pointed out, this tracker takes the entire trajectory $\tau = (s_0^{\text{obs}}, \pi_0, r_0^{\text{soc}}, \dots)$ as input and infers the most likely latent traits that explain the observed agent behaviors throughout the episode.
>
> We agree that defining this explicitly before its first appearance in Equation (3) would significantly enhance the paper's readability. In the revised manuscript, we will move the detailed description of the Posterior Trait Tracker and the definition of $\hat{m}_{\text{post}}$  to appear just before Equation (3), ensuring the notation is clear before being used in the objective function.
>
> >**Response to  Weakness 2 & Question 2**:
>
> As you correctly anticipated, the primary advantage of our SWM-AP framework lies in handling heterogeneity, so we would not expect a significant performance gain in a homogeneous scenario. The most important outcome, as you noted, is to ensure that our method does not underperform the baselines in terms of asymptotic reward. We have conducted this experiment in the Facility Location setting by assigning all agents the identical ground-truth trait. Our results confirm your hypothesis:
>
> MBPO's converged reward of 0.89 ± 0.05. SWM-AP achieves a final performance level statistically on par with the MBPO baseline(0.87 ± 0.04). MBPO reached its final performance level (a reward of 0.89) at approximately 269k training steps, while SWM-AP reached this performance threshold (a reward of 0.89) at around 301k steps. The sample efficiency of SWM-AP is slightly lower than MBPO in this setting, as the model initially expends capacity to model non-existent trait variations. However, the asymptotic performance of SWM-AP converges to a level that is statistically on par with the MBPO baselines.
>
> This demonstrates that our framework is robust; while it is specialized for heterogeneous systems, it does not suffer a significant performance degradation in simpler, homogeneous environments. We will add a new subsection in the appendix of the revised paper to present these results and the corresponding learning curves.
>
> >**Response to Question 3**:
>
> Thank you for raising this critical point regarding error accumulation in model-based methods. This is indeed a fundamental challenge. Our approach mitigates this issue, primarily due to the hierarchical decision-making structure and time-scale separation that are central to our mechanism design formulation.
>
> In our framework, the mechanism design policy (the "Principal") operates at a much slower time scale than the background agents. For instance, in the AI-Economist experiment, the Principal updates the tax policy only once every 100 agent-level steps. This means that for the principal's policy optimization, the effective "horizon" it needs to look ahead is naturally bounded. To evaluate a potential policy change, our SWM only needs to simulate the system's evolution over this relatively short, high-level decision interval . This structure naturally limits the required rollout length for the Principal's policy learning, thereby preventing the catastrophic accumulation of prediction errors that plagues methods requiring very long-horizon rollouts.
>
> This structural advantage is complemented by the higher intrinsic accuracy of our SWM. By conditioning on inferred agent traits, our model achieves a lower single-step prediction error (as shown in Figures 3b and 4b), which further enhances the reliability of these necessary short-to-medium-horizon simulations.
>
> For future work on problems that might require even longer-term strategic planning by the Principal, we agree that more advanced techniques would be beneficial. Methods such as incorporating a learned terminal value function to bootstrap long-term reward estimates, or leveraging more sophisticated trajectory optimization techniques, would be promising directions to further enhance robustness. We will add a brief note on this to the discussion section in our paper.
>
> >**Response to Question 4**:
>
> Thank you for raising this important point. We have extended the training to 5e8 steps and observed that the state loss for our proposed method (SWM) stabilizes at **0.72 ± 0.02**, remaining consistently lower than the MBPO baseline, which achieves **0.83 ± 0.03**. We will include the extended results and updated figure in the camera-ready version to provide a clearer illustration of the training dynamics and confirm that the loss does not diverge further.

---

> > ### Comment · Reviewer_ggqb · 2025-08-04
> >
> > Thank you for clarifying these points, I believe this addresses the main concerns I had about the paper.

---

> ### Author Response · Authors · 2025-08-05
> **Official Comment by Authors**
>
> Thank you very much for taking the time to review our rebuttal. We are glad to hear that it has addressed your main concerns, and we truly appreciate your valuable suggestions, which have helped us significantly improve the manuscript

---

### Note · Authors · 2025-08-13

To the Area Chair and Reviewers,

We are sincerely grateful for your thorough and insightful feedback. The engaging discussion period has been incredibly valuable and has allowed us to significantly enhance the quality and rigor of our manuscript.

We are pleased that our responses and new experiments have addressed the main concerns raised. Specifically, we conducted new scalability experiments with 32 agents, benchmarking against three baselines (PPO, MBPO, Dreamer) across multiple seeds to ensure statistical robustness. These results, along with the runtime and memory footprint analyses, demonstrate our framework's efficiency and scalability. The discussion also encouraged us to explore trait interpretability more deeply, leading to a new diagnostic experiment that validates our model’s ability to learn meaningful traits even under ambiguity.

We also deeply appreciate the forward-looking suggestions. In the final manuscript, we have outlined a roadmap for future work, including scaling our methods to massive systems via techniques like clustering and validating our framework against established behavioral models. These steps are crucial for advancing toward real-world applications.

We have incorporated all clarifications related to notation and experimental design. The experimental results and relevant discussions introduced during the rebuttal will also be included in the final manuscript. We are confident that, thanks to your guidance, the final manuscript will be stronger, clearer, and more impactful.

Thank you once again for your positive engagement and for helping us improve our work.

---

### Decision · Program_Chairs · 2025-09-17

**Decision:**

Accept (poster)

**Comment:**

The paper proposes a hierarchical model-based RL framework for mechanism design in multi-agent systems with latent traits (like skills or preferences). Motivation is clear, the method is theoretically grounded, and evaluation is convincing.

The discussion period has been useful and has helped the authors to improve their work. In particular, they improved evaluation to demonstrate the efficiency and scalability of the proposed framework and have committed to include many clarifications about the experiments in the final version of their paper.

I recommend acceptance because the paper tackles an important and challenging problem, the proposed method is sound both in terms of theoretically motivated and empirical evaluation, and the paper has interesting ideas that might inspire other communities, like those working on agentic frameworks.